# Model-Based Real Time Operation of the Freeze-Drying Process

**Carlos Vilas [1],\*** , **Antonio A. Alonso [1]**, **Eva Balsa-Canto [1]**, **Estefanía López-Quiroga [2] and Ioan Cristian Trelea [3]**

[1] (Bio)Process Engineering Group, Instituto de Investigaciones Marinas (CSIC), Eduardo Cabello, 6. 36208 Vigo, Spain; antonio@iim.csic.es (A.A.A.); ebalsa@iim.csic.es (E.B.-C.)

[2] School of Chemical Engineering, University of Birmingham, Edgbaston, Birmingham B152TT, UK; e.lopez-quiroga@bham.ac.uk

[3] Université Paris-Saclay, INRAE, AgroParisTech, UMR SayFood, F-78850, Thiverval-Grignon, France; cristian.trelea@agroparistech.fr

\* Correspondence: carlosvf@iim.csic.es; Tel.: +34-986-231930

**Abstract:** Background: Freeze-drying or lyophilization is a dehydration process employed in high added-value food and biochemical goods. It helps to maintain product organoleptic and nutritional properties. The proper handling of the product temperature during the operation is critical to preserve quality and to reduce the process duration. Methods: Mathematical models are useful tools that can be used to design optimal policies that minimize production costs while keeping product quality. In this work, we derive an operational mathematical model to describe product quality and stability during the freeze-drying process. Model identification techniques are used to provide the model with predictive capabilities. Then, the model is used to design optimal control policies that minimize process time. Results and conclusion: Experimental measurements suggest splitting the process into two subsystems, product and chamber, to facilitate the calibration task. Both models are successfully validated using experimental data. Optimally designed control profiles are able to reduce the process duration by around 30% as compared with standard policies. The optimization task is introduced into a real time scheme to take into account unexpected process disturbances and model/plant mismatch. The implementation of the real time optimization scheme shows that this approach is able to compensate for such disturbances.

**Keywords:** freeze-drying; operational model; model calibration; real time optimization

## 1. Introduction

Freeze-drying is a preservation technique widely used in pharmaceutical and food processing applications [1,2]. In freeze-drying, products are first frozen and then dehydrated by sublimation of the ice formed during freezing (primary drying) and desorption of the unfrozen water (secondary drying) [3,4]. Such particular processing conditions are key to delivering high-quality, shelf-stable dried products. For example, low processing temperatures avoid the degradation of heat-sensitive compounds (e.g., pharmaceutical active principles or nutrients and flavors in foods), while freezing and subsequent sublimation create highly porous microstructures that enhance rehydration/dissolution of food [5–7] and drug powders [8]. Characteristic low moisture contents make freeze-drying both time-consuming—cycles are typically over 24 h—and energy intensive [9,10]. The challenge here lies in shortening processing times without compromising the quality of the final product—for example, higher temperatures will increase drying rates, but could cause matrix collapse and/or biodegradation.

In this context, optimization protocols based on mechanistic, first-principles models [11–13] can help to achieve the desired time reductions while keeping quality, and also provide a robust tool to deal

with process and/or product disturbances during freeze drying. In recent years, the pharmaceutical sector, operating under strict regulations, has adopted a Quality by Design (QbD) approach that looks for product quality in-line, using model-based optimization strategies combined with process analytical tools (PAT)—temperature and pressure sensors providing dynamic information to estimate critical product and process parameters (e.g., heat and mass transfer resistances in both dried product and vials) [14–19].

However, such approaches are not yet exploited in freeze-drying of foodstuff, in part due to the lack of operational models suitable for real time applications—i.e., current literature in food systems [20–23] does not include simple yet physically meaningful dynamic freeze-drying models—and in part because foods are complex biomaterials, with properties strongly dependent on temperature and pressure (e.g., processing conditions) that are difficult to characterize [13]. Previous works on food systems are scarce, and use either lumped inventory models [3,24] that miss temperature/pressure dependences in the system or very detailed, multiscale models that result computationally involved and include a large number of unknown parameters [25].

To fill this gap, we build on previous work and use a physics-based, operational model for freeze-drying [26] (i.e., reduced attending to separation of time scales) to develop a robust two-layer scheme for real time monitoring, control and optimization of freeze-drying in food matrices. This scheme uses input-output data recorded with standard sensing devices (thermocouples and Pirani gauge) to identify model parameters, such thermo physical properties of the food matrix or characteristic transfer coefficients related to the equipment, that vary with operating conditions (e.g., shelf temperature, chamber pressure). The outputs include Pirani [27] and total chamber pressure, as well as product temperature at the (approximated) bottom of the sample. The inputs are both shelf and condenser temperatures, which act as controls for the system. Once both product and equipment are characterized, the scheme can be used to determine sublimation front position and process end point, to update operating conditions under process disturbances or to design optimal operating policies—off-line and in real time (RTO)—that minimize processing times while ensuring final food quality (i.e., reaching target moisture content while avoiding structure collapse). We assume that, at the beginning of the drying process, the product is frozen and its temperature is uniform in the spatial domain. Therefore, focus of this study is on the drying phase of the process.

Overall, the contribution of this work is two-fold:

1. It shows how the identification problem (i.e., characterization of product and equipment) can be simplified by solving a (decoupled) sequence of parameter identification problems, this is, by considering food matrix and freeze-drying chamber as two separate subsystems, which are described by physics-based operational models developed to satisfy identifiability.
2. It provides a robust, digital tool that supports decision-making in real time for complex food structuring processes (such freeze-drying). This has potential for significant reductions in cost and processing times, waste generation (e.g., reducing batch rejection) and energy demand (e.g., off-line and in-line strategies can be used to minimize energy use during processing) [10,28].

To illustrate the capabilities of this approach, we have considered Lactic Acid Bacteria (LAB) as food matrix exemplar. This is a culture starter used in the manufacture of foods (e.g., yogurt, cheese, fermented meats, and vegetables) and probiotic products that is commonly freeze-dried to preserve its functionality [29].

## 2. Lyophilization Plant and Mathematical Model

### 2.1. Lyophilization Pilot Plant and Experimental Setup Description

The experiments required to perform the different tasks in this work were carried out on a LyoBeta special freeze-dryer (Telstar, Terrassa, Spain) equipped with three thermocouples, a capacitive manometer and a Pirani gauge.

A schematic representation of the pilot plant is shown in Figure 1. The product is placed over different shelves. Such shelves are connected to a heating system which is used to heat and cool down the product. As the product is heated, ice begins to sublimate starting by the top of the product. At this moment two phases can be distinguished, a dried one at the top of the product and a frozen one at the bottom of the product. As the process evolves the frozen region reduces its size till it disappears. Vapor generated by ice sublimation is removed by the action of a condenser to control the pressure in the chamber.

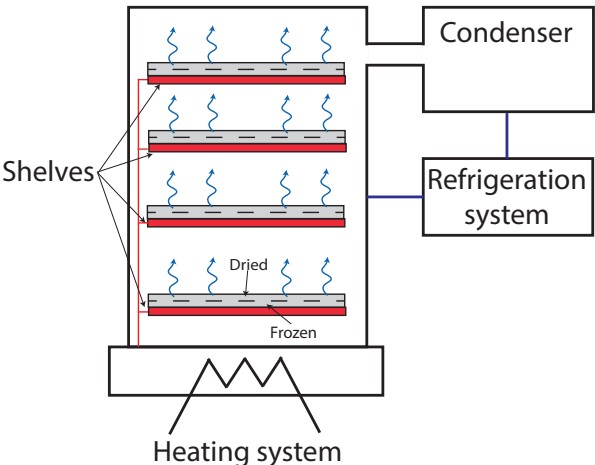

**Figure 1.** Schematic representation of the lyophilization pilot plant.

Lactic acid bacteria were produced by fermentation in controlled conditions of pH and temperature [29]. After concentration, the bacterial cells were re-suspended in a 1:2 cells/protective medium ratio. The protective medium was composed of 200 g $L^{-1}$ of sucrose and 0.15 M of NaCl. A stainless steel tray was filled with 450 g of bacterial suspension.

### 2.2. A Physical Model for the Freeze-Drying Process

The freeze-drying process mainly consists of the following three stages [30]:

- Freezing. The product is frozen in controlled conditions to avoid possible damage, by crystal growth, to the food or biological material. Our model does not include this stage. The product temperature is assumed to be uniform at the end of the freezing stage, and it is used as the initial temperature for the primary drying.
- Primary drying. In this stage, ice is removed from the product by sublimation. Pressure conditions are kept below the triple point and the product is heated from the bottom. An excessive temperature increase in this stage will cause product to collapse so it must be kept below a given value, see Section 2.2.4 for details.
- Secondary drying. The aim is to remove water bound to the solid matrix by desorption. This stage allows for reaching low moisture contents. The food product is more stable during secondary drying so its temperature may be increased to accelerate the process.

Accurate models describing the primary drying stage are computationally involved as this stage involves complex heat and mass transfer mechanisms, as well as ice sublimation. Besides, such models usually include a large number of unknown parameters which often poses identifiability problems when trying to estimate parameter values from experimental data. Operational models with good predictive capabilities can be obtained by considering the following, widely employed, assumptions [26,31,32]:

- The frozen region has uniform heat and mass transfer properties.

- The interface between the frozen and dried layers (sublimation front) is continuous and has infinitesimal thickness.
- Vapor and ice are at equilibrium at the interface.
- The matrix pore structure is permeable to the vapor flux and it is not deformable.

Model complexity can be further reduced by focusing on the phenomena of interest from the process control point of view and neglecting mechanisms occurring at time scales very different from such phenomena. The authors in [26] performed a careful dimensionless analysis revealing several time-scales involved during primary drying. Such analysis is used in this work to obtain the final model equations.

The following section presents an operational version of the freeze-drying model. The parameters involved in the model are summarized in Table 1.

**Table 1.** Parameters involved in the freeze-drying model. Parameters to be estimated are indicated in the "Value" column as t.b.e.

| Parameter | Value | Units | Description |
|---|---|---|---|
| $\rho_{dr}$ | 200.31 | $\mathrm{kg\,m^{-3}}$ | Dried region density |
| $\rho_{fr}$ | 1001.6 | $\mathrm{kg\,m^{-3}}$ | Frozen region density |
| $c_{p,dr}$ | 1254 | $\mathrm{J\,kg^{-1}\,K^{-1}}$ | Dried region heat capacity |
| $c_{p,fr}$ | 1818.8 | $\mathrm{J\,kg^{-1}\,K^{-1}}$ | Frozen region heat capacity |
| $\kappa_{dr}$ | t.b.e. | $\mathrm{W\,m^{-1}\,K^{-1}}$ | Dried region heat conductivity |
| $\kappa_{fr}$ | 2.4 | $\mathrm{W\,m^{-1}\,K^{-1}}$ | Frozen region heat conductivity |
| $\sigma$ | $5.6704 \times 10^{-8}$ | $\mathrm{W\,m^{-2}\,K^{-4}}$ | Stefan–Boltzmann constant |
| $e_p$ | 0.78 | - | Thermal emissivity at the product top |
| $f_p$ | 0.99 | - | Geometrical correction factor |
| $K_{clap}$ | $1.6548 \times 10^{-4}$ | $\mathrm{K^{-1}}$ | Constant in the Clapeyron equation |
| $\Delta H_s$ | $2791.2 \times 10^{-3}$ | $\mathrm{J\,kg^{-1}}$ | Sublimation heat |
| $R$ | 8314 | $\mathrm{Pa\,m^3\,K^{-1}\,kmol^{-1}}$ | Ideal gas constant |
| $L_x$ | $5.75 \times 10^{-3}$ | m | Food product height |
| $L_z$ | 0.242 | m | Food product length |
| $L_y$ | 0.307 | m | Food product width |
| $M_w$ | 18 | $\mathrm{kg\,kmol^{-1}}$ | Water molecular mass |
| $h_{L,1}$ | 3.3 | $\mathrm{W\,m^{-2}\,K^{-1}}$ | Heat transfer coefficient constant |
| $h_{L,2}$ | t.b.e. | $\mathrm{Pa^{-1}}$ | Heat transfer coefficient constant |
| $h_{L,3}$ | 34.4 | Pa | Heat transfer coefficient constant |
| $k_1$ | 430.0 | $\mathrm{s\,m\,kg^{-1}}$ | Mass transfer coefficient constant in Darcy's equation |
| $k_2$ | t.b.e. | $\mathrm{s\,Pa\,kg^{-1}}$ | Mass transfer coefficient constant in Darcy's equation |
| $\beta$ | t.b.e. | $\mathrm{kg\,s^{-1}\,K^{-1}}$ | Mass transfer coefficient constant in chamber/condenser flux |
| $T_{ch}$ | 293.15 | K | Chamber temperature |
| $V_{ch}$ | 0.202 | $\mathrm{m^3}$ | Chamber volume |
| $\tau_A^{ref}$ | $2.689 \times 10^4$ | s | Compartment A reference time constant |
| $\tau_B^{ref}$ | $6.493 \times 10^5$ | s | Compartment B reference time constant |
| $E_a$ | $4.271 \times 10^4$ | $\mathrm{kJ\,kg^{-1}}$ | Activation energy in desorption model |
| $T_{ref}$ | 273.15 | K | Reference temperature in desorption model |
| $\alpha_A$ | 0.669 | - | Compartment A ratio between equilibrium water content |
| $\alpha_B$ | 0.331 | - | Compartment B ratio between equilibrium water content |
| $M_g$ | 0.0434 | $\mathrm{kg\text{-}water\,kg^{-1}\text{-}total}$ | Constant of the GAB equation |
| $C_g$ | 7.4789 | - | Constant of the GAB equation |
| $K_g$ | 0.9827 | - | Constant of the GAB equation |
| $K_{T,g}$ | 8.2 | - | Constant of the glass transition temperature |
| $T_{g,l}$ | −135 | °C | Constant of the glass transition temperature |
| $T_{g,s}$ | 75.58 | °C | Constant of the glass transition temperature |
| $\alpha$ | 1.6 | - | Ratio of molecular heat conductivities of vapor and nitrogen |

### 2.2.1. The Primary Drying

The phenomenon of interest, from the process control point of view, is the heat transport in the product. Based on the time scale analysis, the authors in [26] concluded that vapor flux contribution to the energy balance can be neglected as it is much faster than heat conduction in the solid matrix. Besides, product dimensions $L_z$ and $L_y$ are two orders of magnitude larger than $L_x$, see Table 1 and Figure 1. Therefore, the 3D problem can be accurately approximated by a 1D in the $L_x$ dimension. Under these conditions, the energy balance leads to the following set of partial differential equations (see [26] for details):

$$\rho_{dr} c_{p,dr} \frac{\partial T_{dr}}{\partial t} = \kappa_{dr} \frac{\partial^2 T_{dr}}{\partial \xi^2} \tag{1}$$

$$\rho_{fr} c_{p,fr} \frac{\partial T_{fr}}{\partial t} = \kappa_{fr} \frac{\partial^2 T_{fr}}{\partial \xi^2} \tag{2}$$

$T_i(t, \xi)$, with $i = dr, fr$, represents the food product temperature; $\xi$ are the spatial coordinates (product height); and $dr, fr$ denote dried and frozen regions, respectively.

Note that, during primary drying, sublimation makes the front to move from the top to the bottom of the product. Front velocity $w = \frac{dx}{dt}$, where $x(t)$ denotes the front position, is computed using the Stefan condition [32]:

$$(\rho_{fr} - \rho_{dr}) \Delta H_s w = \left( \kappa_{fr} \left. \frac{\partial T_{fr}}{\partial \xi} \right|_{\xi=x^+} - \kappa_{dr} \left. \frac{\partial T_{dr}}{\partial \xi} \right|_{\xi=x^-} \right) \tag{3}$$

where $\Delta H_s$ is the sublimation heat.

At the product top ($\xi = 0$), radiation is considered as the main heat transport phenomena, hence the heat flux at this boundary reads as:

$$\kappa_{dr} \left. \frac{\partial T_{dr}}{\partial \xi} \right|_{\xi=0} = \sigma e_p f_p (T_{ch}^4 - T_{dr}|_{\xi=0}^4) \tag{4}$$

where $\sigma$ is the well-known Stefan–Boltzmann constant, $e_p$ is the emissivity, $f_p$ is a geometric correction factor and $T_{ch}$ is the chamber temperature. At the product bottom ($\xi = L$), we consider both radiation and conduction via gas-surface collisions [33]:

$$\kappa_{fr} \left. \frac{\partial T_{fr}}{\partial \xi} \right|_{\xi=L} = h_L (T_{sh} - T_{fr}|_{\xi=L}) \tag{5}$$

where $T_{sh}$ is the shelf temperature. Heat transfer coefficient $h_L$ is computed as a function of the chamber pressure, $P_{ch}$, (see [33] for details):

$$h_L = h_{L,1} + \frac{h_{L,2} P_{ch}}{1 + \dfrac{P_{ch}}{h_{L,3}}} \tag{6}$$

At the front ($\xi = x$), continuity of temperature across the front is considered, i.e.,

$$T_{dr}|_{\xi=x} = T_{fr}|_{\xi=x} = T_{front} \tag{7}$$

At this boundary, vapor and ice are assumed to be in equilibrium so that the front temperature ($T_{front}$) can be computed from the front pressure using the Clausius–Clapeyron equation:

$$T_{front} = \frac{1}{\dfrac{1}{273.11} - K_{clap} \log \left( \dfrac{P_{front}}{611.72} \right)} \tag{8}$$

We use Darcy's law, which states that the vapor flux per unit area is proportional to the pressure gradient, to compute the front pressure ($P_{front}$):

$$\frac{\partial P}{\partial \xi} = \frac{\mu}{K} \frac{w(\rho_{fr} - \rho_{dr})}{\rho_v} \tag{9}$$

where $K$ corresponds with the permeability of the porous matrix; $\mu$ represents the vapor viscosity and $\rho_v$ the vapor density. It has been experimentally observed that the mass transfer coefficient in Equation (9) has a linear dependency with the thickness of the dried layer ($x$). Furthermore, in many freeze-dried products there exists a mass resistance for zero layer thickness [33]. Such resistance, which depends on the chamber pressure, can be attributed to a crust formed on the product or the pore creation process at the sublimation front and should be included in the mass transfer coefficient expression. Taking these issues into account, Equation (9) results into:

$$\frac{\partial P}{\partial \xi} = K_c w(\rho_{fr} - \rho_{dr}) \tag{10}$$

where

$$K_c = k_1 P_{ch} + k_2 x \tag{11}$$

with $k_1$ and $k_2$ being given parameters.

Advanced numerical techniques, like the Arbitrary Lagrangian-Eulerian (ALE) algorithm [32], are usually employed to solve the moving front problem. In general, such techniques require the use of specialized software. We will apply the Landau transform [34,35] to overcome such limitation. As a result, we will obtain an equivalent system with fixed domain in which classical numerical techniques, such as the finite element method, can be applied. See Appendix A for details.

### 2.2.2. The Secondary Drying

As mentioned before, at this stage of the process there is no frozen water and the bounded water is eliminated by desorption. Since there is only one region (dried region), there is no moving boundary and classical numerical methods can be employed to solve the model equations. Equation (1) with the boundary conditions defined by Equations (4) and (5) , where $\xi \in [0, L]$, is used to describe product temperature evolution and distribution during this stage.

### 2.2.3. The Condenser Model

Part of the vapor produced as a result of ice sublimation is removed from the plant by the action of a condenser. The evolution of the chamber pressure ($P_{ch}^v$) is accurately described by the general gas equation, since working pressures in the chamber are low:

$$\frac{dP_{ch}^v}{dt} = \frac{RT_{ch}}{M_w V_{ch}} \frac{dm_{ch}^v}{dt}$$

$R$ is the ideal gas constant, $M_w$ denotes the molecular weight of water and $V_{ch}$ is the chamber volume. Vapor accumulated in the chamber can be obtained through a mass balance between the chamber and the condenser:

$$\frac{dm_{ch}^v}{dt} = \phi_p^v - \phi_c^v \tag{12}$$

where $\phi_p^v$ and $\phi_c^v$ denote, respectively, the flux of vapor from the product to the chamber and from the chamber to the condenser. Flux $\phi_p^v$ corresponds with the flux of ice being sublimated, i.e.,:

$$\phi_p^v = (\rho_{fr} - \rho_{dr}) L_z L_y \frac{dx}{dt} \tag{13}$$

with $L_z$ and $L_y$ being, respectively, the length and width of the product sample.

The second term on the RHS of Equation (12) ($\phi_c^v$) can be described using the mass transfer theory in binary gas mixtures, including bulk flow and mutual diffusion terms [36]:

$$\phi_c^v = \frac{1}{\beta T_{ch}} \ln \left( \frac{P_{ch} - P_c^v}{P_{ch} - P_{ch}^v} \right) \tag{14}$$

where $\beta = \frac{l_{eff} R}{L_z L_y M_w} P_{ch} D_{vN}$. $D_{vN}$ and $l_{eff}$ are the mutual diffusion coefficient and the effective duct length, respectively. The details regarding the derivation of Equation (14) are presented in [36].

Note that Equation (13) can only be used during primary drying where a sublimation front, that evolves with velocity $w = \frac{dx}{dt}$, exists. During secondary drying, flux $\phi_p^v$ is assumed to be zero. In the real process, the heat coming trough the edges of the shelf accelerate the sublimation in those parts of the product that are closer to the borders. As a consequence, vapor flux and vapor pressure decrease gradually in the transition between primary and secondary drying. To describe this phenomenon, 2D or 3D models are required [37]. Such models are unsuitable for RTO purposes as they require long simulation times. Alternatively, 1D models, in which the transition between primary and secondary drying is instantaneous, are operational and yet accurate to describe systems where one of the spatial dimensions is much smaller than the other two [26,31,38,39], as it is the case in the present study. Besides, chamber pressure in the transition between primary and secondary drying has little effect on the desorption kinetics [40,41] and, therefore, on the product quality.

### 2.2.4. The Desorption Model

During primary drying, ice sublimation and desorption of bound water are the physical mechanisms for water loss. In the secondary drying stage, no ice remains in the product so desorption is the only mechanism for water loss.

The desorption model will be used to determine final water content in the product, which is related to the final product quality, as well as the glass transition temperature, which is related to product stability during the process. This variable is used to define constraints on the foodstuff temperature that prevents product collapse.

The initially present water is distributed among two "compartments" (compartments A and B), corresponding to different physical states and/or interactions with the solid matrix (bacterial cell, membrane, cryoprotectant, etc.), which have their own temperature-dependent drying kinetics [40]:

$$\frac{dc_{w,i}}{dt} = -\frac{1}{\tau_i} \left( c_{w,i} - c_{w,i}^{eq} \right); \quad \text{with } i = A, B \tag{15}$$

with $c_{w,i}^{eq}$ being the equilibrium water content for compartment $i$ at equilibrium. $\tau_i$, desorption time functions, are assumed to obey an Arrhenius-like relationship [41]:

$$\tau_i = \tau_i^{ref} \exp \left[ \frac{E_a}{R} \left( \frac{1}{T_p} - \frac{1}{T^{ref}} \right) \right]$$

where $\tau_i^{ref}$ is the value of the time constant at an arbitrarily fixed reference temperature and $E_a$ is the activation energy. Equilibrium water content for compartment $i$ is related to the total equilibrium water content as:

$$c_{w,i}^{eq} = \alpha_i c_w^{eq}; \quad \text{with } \alpha_A + \alpha_B = 1$$

$c_w^{eq}$ is computed using the water activity ($a_w$) based on the classical Guggenheim-Anderson-Boer (GAB) equation for the sorption isotherm [29,42]:

$$c_w^{eq} = \frac{M_g C_g K_g a_w}{(1 - K_g a_w)(1 + K_g a_w (C_g - 1))} \tag{16}$$

where $M_g$, $K_g$ and $C_g$ are given, product dependent, constants. Water activity is the ratio between the chamber vapor pressure and the equilibrium vapor pressure, i.e., $a_w = P_{ch}^v / P_{eq}^v$ where $P_{eq}^v$ is related to product temperature as [43,44]:

$$P_{eq}^v = 3.5941 \times 10^{12} \exp\left(-\frac{6144.96}{T_p}\right) \tag{17}$$

which is valid for ice and:

$$P_{eq}^v = 8.4755 \times 10^{11} \exp\left(-\frac{5750.38}{T_p}\right)\left(1 - 0.0045(T_p - 273.15)\right) \tag{18}$$

whenever water is involved.

The glass transition temperature is related to the water content as follows [29]:

$$T_g(c_w) = \frac{K_{T_g} T_{g,1} c_w + (1 - c_w) T_{g,s}}{K_{T_g} c_w + (1 - c_w)} \tag{19}$$

where $K_{T_g}$, $T_{g,1}$ and $T_{g,s}$ are given parameters.

As mentioned above, $T_g$ can be used to predict product integrity. In this regard, product temperature should not exceed $T_g$ since foodstuff might collapse.

## 3. Strategies for Model Parameter Identification

The aim of parameter estimation is to find the values of the unknown model parameters that minimize the distance between model simulations and experimental data. To that purpose, the following elements are required:

- A set of model parameters to be estimated. Usually, these are parameters that are difficult to measure or cannot be found in the literature.
- Experimental measurements to compare against model results. Measured variables are usually state variables (e.g., temperature, concentration, pressure) or combinations of state variables. The quantities to be measured are known as the observables. Changes on the unknown model parameters should have an effect on the observables, otherwise such parameters cannot be estimated.
- A measure of the distance between model predictions and experimental data (cost function).
- An optimization algorithm to find the parameter values that minimize the cost function.

Let us denote by $\theta$ the set of unknown model parameters and by $y$ the observable. In this work, the cost function is defined as the Root Mean Square Error (RMSE) value. The parameter estimation problem is formulated as, find:

$$\min_\theta (RMSE); \quad \text{with} \quad RMSE = \sqrt{\frac{\sum_{k=1}^{n_e} \sum_{i=1}^{n_{s,k}} (y_{meas}(t_{i,k}) - \bar{y}_{mod}(t_{i,k}))^2}{\sum_{k=1}^{n_e} n_{s,k}}} \tag{20}$$

subject to the model dynamics presented in previous section. In Equation (20), $n_e$ is the number of experiments for estimation purposes, $n_{s,k}$ is the number of sampling times in experiment $k$ and $\bar{y}_{mod}(t_{i,k})$ represents the model prediction of the observable at time $t_i$ in experiment $k$.

The optimization method chosen to solve the problem is the sequential hybrid global-local Enhanced Scatter Search (ESS) [45] which is included in the AMIGO2 toolbox [46].

The freeze-drying system was divided in two subsystems to facilitate the estimation task, the condenser and the product, each of them with an associated mathematical model as shown in the previous section. The procedure is schematically represented in Figure 2.

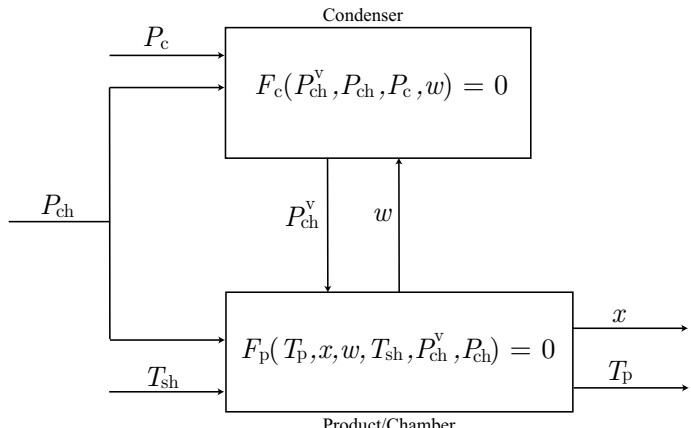

**Figure 2.** Schematic representation of the freeze drying process. Two subsystems are considered, the condenser and the product. The arrows entering the subsystems represent the process control variables whereas output arrows represent the state variables.

The control variables, represented in Figure 2 as arrows entering the subsystems, correspond to those process variables that can be directly manipulated to actuate into the process. Three controls are available in this pilot plant: condenser pressure ($P_c$), chamber pressure ($P_{ch}$) and shelf temperature ($T_{sh}$). These variables are also measured so the real value can be used for parameter estimation tasks. Arrows leaving the subsystems in Figure 2 represent the state variables. The chamber vapor pressure ($P_{ch}^v$) is the only state variable computed in the condenser model. State variables in the product subsystem are the front position ($x$) and velocity ($w$) as well as product temperature ($T_p$), the latter distributed along the product height. Both subsystems, condenser and product, are coupled through two state variables, $P_{ch}^v$ and $w$.

In this system, Pirani pressure ($P^p$) and total chamber pressure ($P_{ch}$) are also measured. Since chamber vapor pressure is related to $P^p$ and $P_{ch}$ through the formula [36]:

$$P_{ch}^v = \frac{P^p - P_{ch}}{\alpha - 1}; \qquad \alpha = 1.6, \tag{21}$$

we can consider that $P_{ch}^v$ is indirectly measured. These indirect measurements can be used, in the product model, so that the condenser model can be neglected in the estimation of the product model parameters. In this regard, the estimation task will proceed in two steps:

- Estimation of the unknown parameters involved in the product model, using the indirect measurements of $P_{ch}^v$ instead of the condenser model. On the one hand, this reduces the number of parameters to be estimated together since the unknown condenser parameter is not taken into account. On the other hand, possible modeling errors on the condenser model are avoided.
- Estimation of the unknown parameters involved in the condenser model. As it is sketched in Figure 2, this estimation requires the product model and the parameters estimated in the previous step to compute the front velocity. Note that, if a procedure to measure $w$, such as an on-line sensor of front velocity, were available, product model could be disregarded in this step.

The following freeze-drying experimental protocol was applied is the different tests:

- Product freezing using $-50\,°C$ as a set point (cooling rate of $0.6\,°C\,\text{min}^{-1}$).
- Primary drying varying from freezing temperature to $-10$, $0$ or $20\,°C$ depending on the experiment.

- Secondary drying using 25 °C as the set point.
- Total chamber pressure was kept at 20 or 60 Pa depending on the experiment.

*3.1. Parameter Estimation in the Product/Chamber Subsystem*

The subset of model parameters to be estimated is $\theta = [\kappa_{dr}, k_2, h_{L,2}]$, i.e., the heat conductivity of the dried region; product/chamber mass resistance coefficient—see Equation (11); and heat transfer coefficient—see Equation (6).

The observable in this subsystem is the temperature at the bottom of the product. Because of limitations on the equipment, the precise location of such thermocouple is not known with precision. Therefore, a mean value of the temperatures in the bottom region of the product is used as the observable $y = T_{p,meas}$. Such region is 1 mm long.

In this subsystem, Equation (20) may be rewritten as:

$$\min_{\kappa_{dr}, k_2, h_{L,2}} (RMSE); \quad \text{with} \quad RMSE = \sqrt{\frac{\sum_{k=1}^{n_e} \sum_{i=1}^{n_{s,k}} \left( T_{p,meas}(t_{i,k}) - \bar{T}_{p,mod}(t_{i,k}) \right)^2}{\sum_{k=1}^{n_e} n_{s,k}}} \quad (22)$$

where $\bar{T}_{p,mod}(t_{i,k})$ represents the mean value of the temperatures in the bottom region of the product.

The experimental data employed in the parameter estimation were obtained from five experiments. Each experiment differs from the others in the control variables, i.e., shelf temperature ($T_{sh}$) and chamber pressure ($P_{ch}$). Besides, although the set point for the freezing temperature was set to $-50$ °C, the refrigerant group was not able to reach such temperatures. Experimental measurements showed that the actual values turn out to range between $-37$ and $-43$ °C, depending on the experiment. Therefore, initial conditions for primary drying also differ from one experiment to another.

A cross validation procedure was applied [47] to robustly estimate the parameters. In this procedure, part of the experimental data is saved for validation purposes, i.e., such data is not used in the estimation task. In this case, we use four of the five experiments for the parameter estimation task, leaving the remaining one for validation. To increase robustness of the results, we have performed five estimations, each one excluding a different experiment.

The product model, together with the measured values of the temperature at the bottom of the product ($T_{p,meas}$) in the experiments considered for estimation, were implemented in the AMIGO2 toolbox. RMSE values were obtained by comparing experimentally measured values of the product temperature with the simulation results.

The estimation results, which include optimal parameter values as well as RMSE values for both estimation and validation tests, are summarized in Table 2.

**Table 2.** Cross validation results for the product model. Five estimations of the parameters were performed, each one excluding a different experiment. $RMSE_e$ and $RMSE_v$ denote, respectively, the Root Mean Square Error (RMSE) values obtained for the estimation and validation experiments.

| Parameter | Excluded Experiment | | | | |
|---|---|---|---|---|---|
| | 1 | 2 | 3 | 4 | 5 |
| $h_{L,2}$ | 1.96 | 1.86 | 1.93 | 1.82 | 2.03 |
| $k_2$ | $8.60 \times 10^7$ | $7.86 \times 10^7$ | $6.52 \times 10^7$ | $9.41 \times 10^7$ | $9.51 \times 10^7$ |
| $\kappa_{dr}$ | $2.45 \times 10^{-4}$ | $4.13 \times 10^{-4}$ | $2.41 \times 10^{-4}$ | $2.85 \times 10^{-4}$ | $2.85 \times 10^{-4}$ |
| $RMSE_e$ | 2.15 | 1.64 | 2.11 | 2.10 | 2.00 |
| $RMSE_v$ | 1.62 | 3.10 | 2.01 | 2.49 | 2.54 |

The values found for the parameters are in accordance with the expected physical range. Besides, variability in the parameter estimates of the cross validation procedure for $h_{L,2}$, $k_2$ and $\kappa_{dr}$ is 4.4%, 14.7% and 23.7%, respectively. While the variability of $\kappa_{dr}$ is much larger than the variability of the other two parameters, the results are considered to be robust.

Figure 3 illustrates the predictive capabilities of the model.

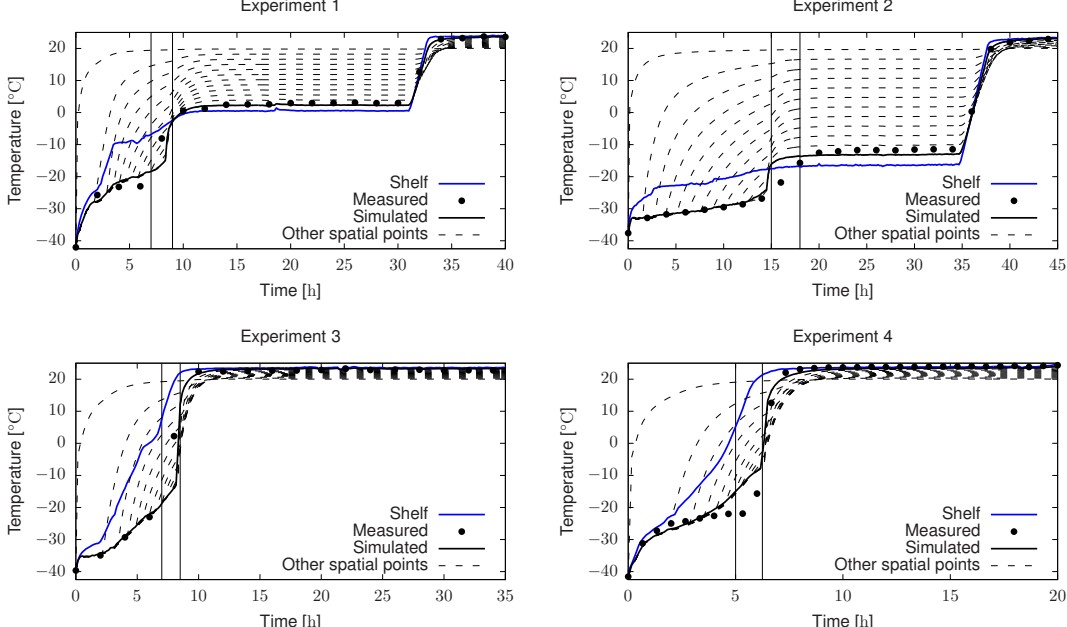

**Figure 3.** Predictive capabilities of the freeze-drying model from the product temperature point of view. Blue line corresponds to the shelf temperature. Continuous black line and black dots represent, respectively, the model predictions and experimental measurements at the bottom of the product. Dashed black lines correspond to the model prediction at other spatial points in the product. Vertical lines represent the end of primary drying and the beginning of secondary drying.

Four subplots are shown, each one corresponding to the validation experiment in the cross validation procedure. Blue line represents the shelf temperature (control variable) evolution in each experiment whereas dashed lines correspond to model predictions of the product temperature evolution at different spatial points. The figure shows that model predictions at the bottom of the product (continuous black line) reproduce the experimental behavior (black dots). In particular, the end of the primary drying stage as well as the product temperature during such stage are predicted by the model with a reasonable degree of accuracy. Temperature during secondary drying is also accurately described by the model. However, some mismatch between predicted and measured temperatures can be observed in the transition from primary to secondary drying (highlighted in Figure 3 by two vertical black lines). In fact, the model seems to predict a faster temperature increase than the one measured by the thermocouple.

So far, no clear explanation has been found to such disagreement although it most probably can be attributed to an interphase structure which is more complex than the sharp transition between a frozen and a dried region as it is considered by the model. Nevertheless, as mentioned on Section 2.2.3, this does not have a significant impact on the product quality. Besides, product temperature predicted by the model in the transition is usually larger than the experimental measurements. Therefore, as it will be described in Section 4.1, operation policies that satisfy safety constraints according to model predictions will also satisfy such constraints in the experiments.

### 3.2. Parameter Estimation for the Condenser Model

The unknown model parameter in the condenser model is coefficient $\theta = \beta$, see Equation (14). In this plant, it is possible to experimentally measure the total chamber and Pirani pressures so that the chamber vapor pressure ($P_{ch}^v$) can be obtained, see Equation (21). $y = P_{ch}^v$ is used as the observable in the estimation task.

In this case, the optimization problem defined by Equation (20) reads as:

$$\min_{\beta}\left(RMSE\right); \quad \text{with} \quad RMSE = \sqrt{\frac{\sum_{k=1}^{n_e}\sum_{i=1}^{n_{s,k}}\left(P_{ch,meas}^v(t_{i,k}) - P_{ch,mod}^v(t_{i,k})\right)^2}{\sum_{k=1}^{n_e}n_{s,k}}} \tag{23}$$

The same experiments used in the previous section are employed here to estimate the condenser parameter $\beta$. In this case we use both the product model, with the parameter values computed in previous section, and the condenser model.

As in previous section, cross validation is used to evaluate the performance of the model. The estimation results, including optimal value of $\beta$ and the RMSE values corresponding to both the estimation and validation experiments, are presented in Table 3.

**Table 3.** Cross validation results for the product/condenser model. Five estimations of parameter $\beta$ were carried out, each one excluding a different experiment. $RMSE_e$ and $RMSE_v$ denote, respectively, the RMSE values obtained for the estimation and validation experiments.

| Parameter | Excluded Experiment | | | | |
|---|---|---|---|---|---|
| | **1** | **2** | **3** | **4** | **5** |
| $\beta$ | $2.31 \times 10^3$ | $2.17 \times 10^3$ | $2.31 \times 10^3$ | $2.37 \times 10^3$ | $2.37 \times 10^3$ |
| $RMSE_e$ | 1.6 | 1.7 | 2.2 | 2.0 | 1.8 |
| $RMSE_v$ | 2.8 | 2.1 | 0.6 | 0.6 | 2.3 |

The estimated values of $\beta$ are within the expected physical range and close to the values presented in the literature [36]. Besides, the variability in the parameter estimates for the different cases in the cross validation is lower than 4% which evidences the estimation robustness.

Model predictive capabilities, in terms of chamber pressure prediction, are illustrated in Figure 4. As in the previous case, the four subplots correspond to the validation experiment in the cross validation tests.

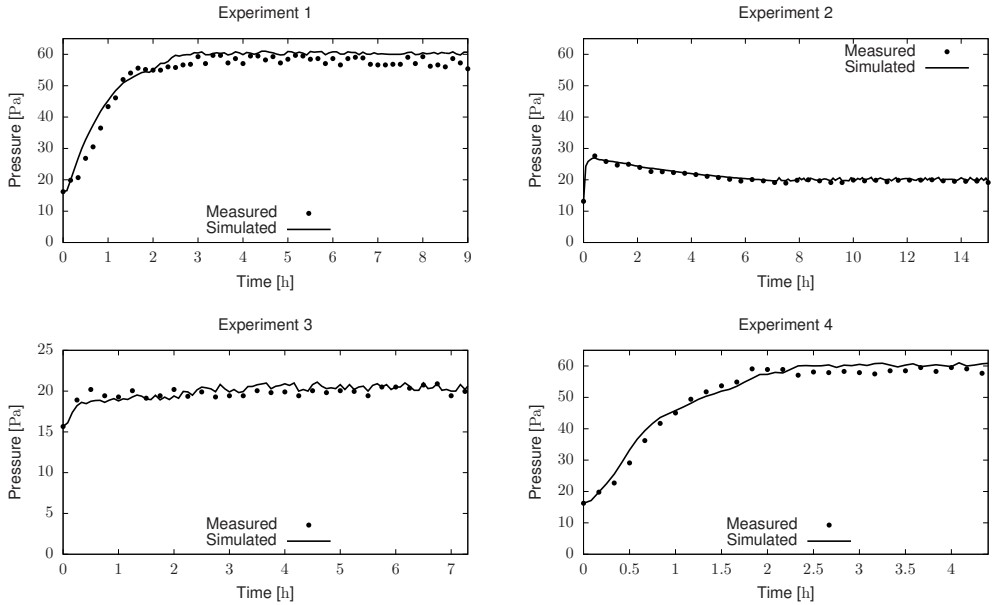

**Figure 4.** Predictive capabilities of the freeze-drying model for the chamber vapor pressure. Black circles represent the chamber vapor pressure obtained from measurements of Pirani and total chamber pressure whereas continuous black lines correspond to the model predictions.

Only primary drying is represented in the figures since during secondary drying vapor pressure is almost zero. As mentioned in Section 2.2.4, the transition between primary and secondary drying is assumed to be instantaneous. Simulation results for the chamber vapor pressure are represented by continuous black lines whereas black circles correspond to experimental measurements. As shown in the figure, the model is able to reproduce the experimental behavior with a high degree of accuracy.

## 4. Real time Optimization of the Freeze-Drying Process

### 4.1. Off-Line Dynamic Optimization

The objective of the dynamic optimization problem is to find the shelf temperature time profile that minimizes the process duration while fulfilling product quality requirements at final time and product temperature constraints during the whole process to avoid product collapse. The other control variable on this system, i.e., chamber pressure, as well as initial conditions for the state variables are taken from the first experiment employed in the parameter estimation task. Mathematically, the optimization problem is stated as:

$$\min_{T_{sh}} t_f \tag{24}$$

$$\text{subject to} \quad -50\,^{\circ}\mathrm{C} \leq T_{sh} \leq 25\,^{\circ}\mathrm{C}$$
$$-\frac{1}{60}\,^{\circ}\mathrm{C}\,\mathrm{s}^{-1} \leq \frac{\mathrm{d}T_{sh}}{\mathrm{d}t} \leq \frac{1}{60}\,^{\circ}\mathrm{C}\,\mathrm{s}^{-1}$$

and to model dynamics, see Section 2.2.

If no other constraints are considered, it seems clear that the optimal profile would be that one corresponding to the maximum shelf temperature allowed, i.e., 25 °C. However, this would result into product collapse and the rejection of the batch. In this way, constraints on the allowed product temperature are defined as function of the glass transition temperature, see Equation (19), to avoid product collapse:

$$T_p(\xi, t) - T_g(\xi, t) \leq 0 \tag{25}$$

In Section 3.1, validation results showed that, in some experiments, the model might overstimate product temperature in the transition between primary and secondary drying. Mathematically, this can be formulated as: $T_{p,meas}(\xi, t_t) < T_{p,mod}(\xi, t_t)$, where $t_t$ represents the transition period time. If the

product temperature predicted by the model satisfies constraint (25), then product temperature in the experiment also satifies it:

$$T_{p,meas}(\xi, t) - T_g(\xi, t) < T_{p,mod}(\xi, t) - T_g(\xi, t) \leq 0 \tag{26}$$

We handle constraints by minimizing the square of the difference, $T_{p,mod}(\xi, t) - T_g(\xi, t)$, in Equation (26) as suggested in [48]. This ensures that condition in Equation (26) is always satisfied.

Product quality is related to the water content at the end of the process. The lower the water content, the better the quality. The following constraint is used to ensure product quality:

$$\int_{\xi} c_w(\xi, t_f) \, d\xi \leq c_{w,max} \tag{27}$$

The value $c_{w,max} = 9.15 \times 10^{-3} \frac{\text{kg water}}{\text{kg total}}$ was taken from the final water content in the first experiment.

There are several alternatives for the solution of dynamic optimization problems [11]. In this case, we have chosen the control vector parameterization (CVP) approach [49], because of its capacity to handle large-scale dynamic optimization problems without solving very large non-linear programming (NLP) problems and without dealing with extra junction constraints [50]. Basically, the control vector parameterization proceeds dividing the process duration into a number of elements and approximating the control functions typically using low order polynomials. The polynomial coefficients become the new decision variables and the solution of the resulting NLP problem (outer iteration) involves the system dynamics simulation (inner iteration).

NLP problems arising from the application of the control vector parameterization method are frequently multimodal (i.e., presenting multiple local optima), due to the highly nonlinear nature of the dynamics [51]. In this work, we will use a method that combine global stochastic methods with local gradient based methods which has proven to perform well for solving NLP problems. The method selected, eSS, is based on the scatter search method, see [45] for details.

In this problem, we have chosen piecewise linear functions, with seven discretization points, to approximate the control in the CVP approach. Figure 5 represents the evolution and spatial distribution of the product temperature (gray lines on the top figures) and water content (bottom figures) as well as the evolution of the shelf temperature (black lines on the top figures).

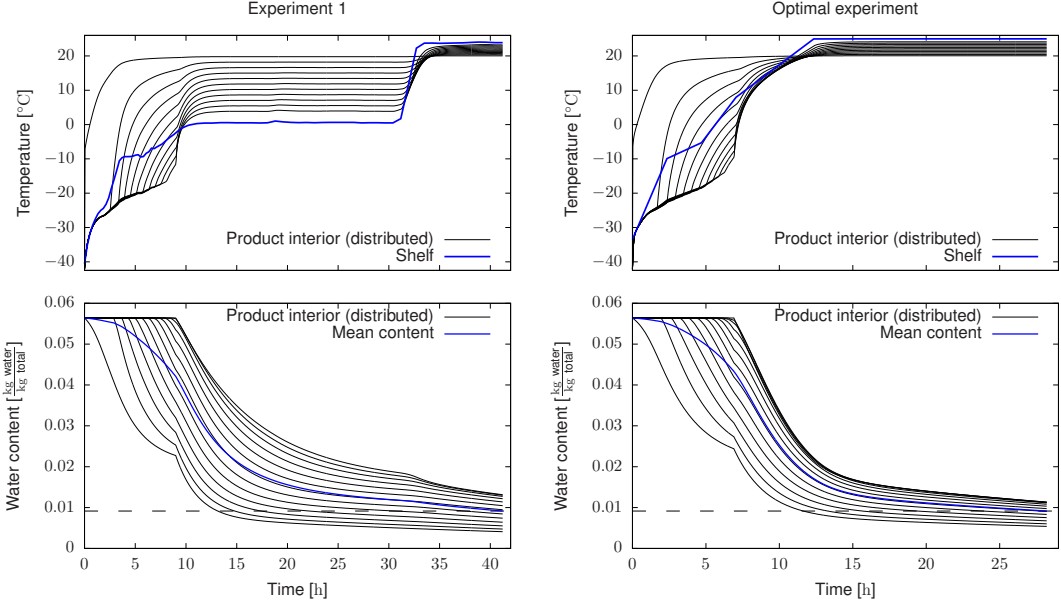

**Figure 5.** Evolution of the product temperature and shelf temperature (top figures) as well as water content (bottom figures). Figures on the left correspond with experiment 1 whereas the optimal design results are represented on the figures on the right. Horizontal line on the bottom figures represent the constraint on the final water content.

As expected, the shelf temperature profile on the optimally designed experiment (see right column subplots) starts at a low value, to avoid product damage, and increases as the product becomes more stable. Note that this profile is more aggressive than the one used in experiment 1 and, as a result, the final time is reduced by a 30% (13 h). Despite the resulting sharp shelf temperature profile, constraints defined in Equations (25)–(27) are fulfilled, i.e., the product integrity is ensured and quality, in terms of final water content, are the same in both experiment 1 and the optimally designed one.

### 4.2. Real Time Optimization

The optimal profile for shelf temperature obtained in previous section was computed assuming that the model represents perfectly the plant behavior and the controller is able to follow precisely the set points. However, in real situations, plant/model mismatch is unavoidable and the PI controller implemented in the freeze-drying plant may be not able to exactly follow the piecewise linear optimal profile. Besides, unexpected plant disturbances, which cannot be taken into account in the off-line profile, may emerge during process operation.

To take into account all these issues, a real time optimization (RTO) scheme is implemented in the plant. The main idea behind the RTO scheme is to periodically use process information to recompute on-line the optimal profiles. The steps of the RTO procedure are the following:

- An optimal profile for the shelf temperature is computed as in previous section.
- Such profile is sent to the freeze-drying plant as the set point for the PI control.
- As the process is running, measurements of relevant variables are being recorded. In this case, we measure, shelf temperature, chamber temperature, Pirani pressure and product temperature.
- After a given period of time, 1 h in this case, measured information is introduced in the model. Then, a new optimal profile is computed by solving the optimization problem defined in Equation (24) and taking into account the new available information.
- Steps 2–4 are repeated till the end of the process.

The objective is to compute the shelf temperature optimal profile that minimizes process time. Process is assumed to be finished when the product mean water content is below $0.0142 \frac{\text{kg Water}}{\text{kg Total}}$.

Chamber pressure set point is fixed to 10 Pa. In this case, bounds on the shelf temperature and its slopes are:

$$-50\,°\mathrm{C} \le T_{sh} \le 40\,°\mathrm{C}; \quad -\frac{1}{120}\,°\mathrm{C\,s}^{-1} \le \frac{\mathrm{d}T_{sh}}{\mathrm{d}t} \le \frac{1}{120}\,°\mathrm{C\,s}^{-1}$$

It must be pointed out that computing optimal policies takes a few minutes. During this time, although the process continues evolving, plant measurements are not taken into account and the RTO scheme assumes that the plant follows the predicted profiles. Unexpected disturbances during this period can make the computed profile non optimal. Such disturbances are, nonetheless, taken into account when the next profile is computed.

Figure 6 illustrates an RTO implementation on the freeze-drying process. Figures at the left hand side, i.e., Figure 6a,c,e, represent shelf temperature and chamber pressure. Figure at the right hand size, i.e., Figure 6b,d,f represent product mean water content. Figure 6a,b correspond with the off-line optimal results, i.e., before the primary drying begins. Therefore, no plant measurements were used in the computation of this profile.

As process evolves, plant measurements are being recorded and new optimal profiles are being recomputed every hour using such information and following the procedure previously indicated. Subplots Figure 6c,d show the situation after 16 h, i.e., around half of the process duration. Blue and red parts of the lines correspond with the measured and recorded data. Black and gray parts of the lines represent the optimal solution from that time. As shown in the figures, the optimal profile has deeply changed. The most important factors for this are:

- Product freezing temperature differs from the expected one, i.e., from the one used in Figure 6a,c. This was because the refrigerant group was not able to reach $-50\,°\mathrm{C}$.
- Initial chamber pressure was around 60 Pa and it was supposed to be 10 Pa. Besides, chamber pressure controller did not perform as expected and most of the time it remains below the set point (10 Pa).
- Shelf temperature controller is not able to exactly follow the optimal profiles.

Once these unexpected disturbances are measured, they are introduced in the RTO scheme which computes a new optimal shelf temperature profile. Such profile allows to reach the quality constraint (water content at the end of the process) without significant increase in process time.

Figure 6e,f show the real (measured) values at the end of the process. Optimal profile has also changed with respect to the situation at 16 h to compensate for plant unexpected disturbances. However, the on-line procedure allowed to reach the desired final product water content.

Finally, we use a simulation experiment to illustrate one of the advantages of the RTO scheme as compared with the off-line optimal control approach (see Figure 7).

To that purpose, we use the experimental measurements shown in Figure 6. After the process has been running for 1 h, i.e., the first time the control profile is updated, we consider two situations:

- We follow the RTO procedure, as in the previous case, and use experimental measurements—shelf temperature (blue segment in Figure 7a) and chamber pressure (red segment in Figure 7a)—to compute a new optimal profile (continuous black line in Figure 7a).
- We do not use available plant measurements and force the system to follow the off-line profile previously computed (dashed black line in Figure 7a).

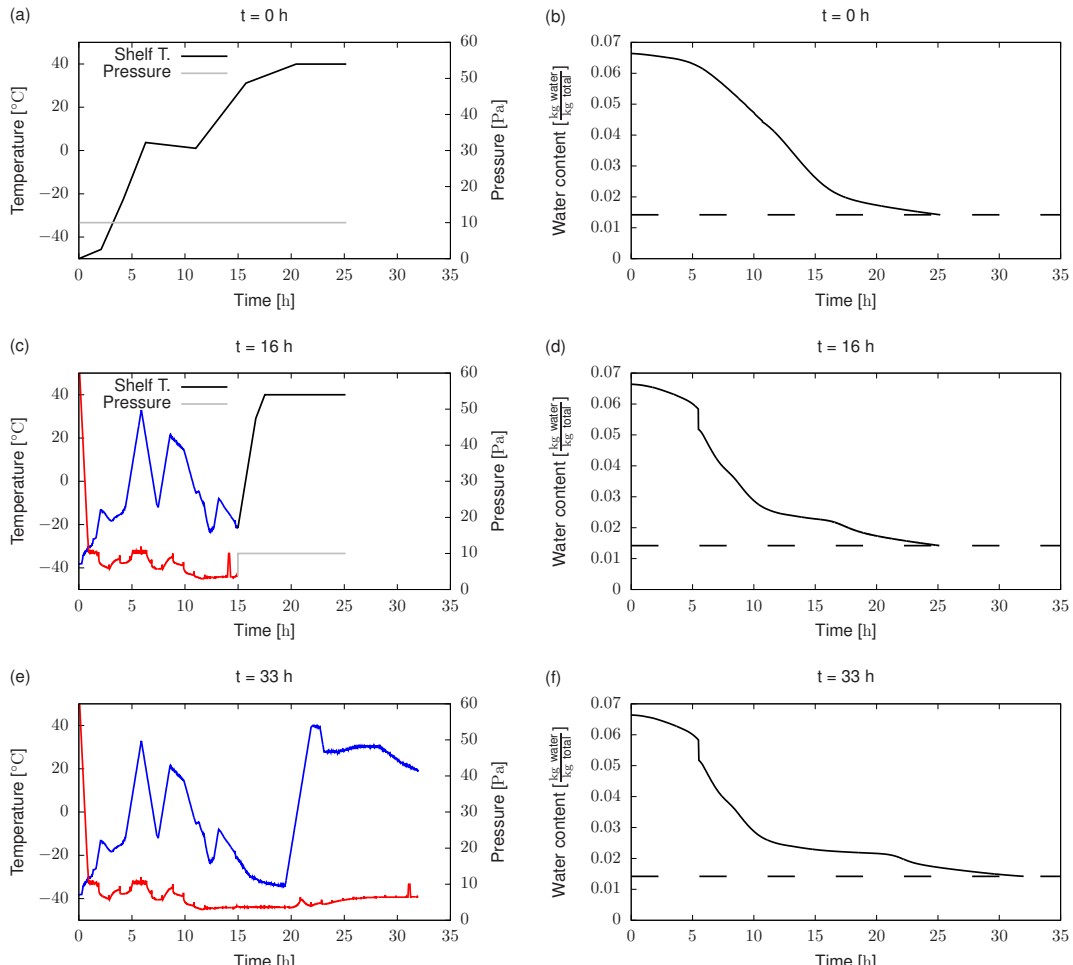

**Figure 6.** Offline and real time optimization (RTO) scheme results. Subfigures (**a,c,e**) represent the evolution of shelf temperature (decision variable) and chamber pressure. Black and gray parts of the profiles correspond with the optimal solution. Blue and red parts correspond with the plant measurements. Subfigures (**b,d,f**) represent the evolution of product mean water content. Dashed line indicates the quality objective to reach.

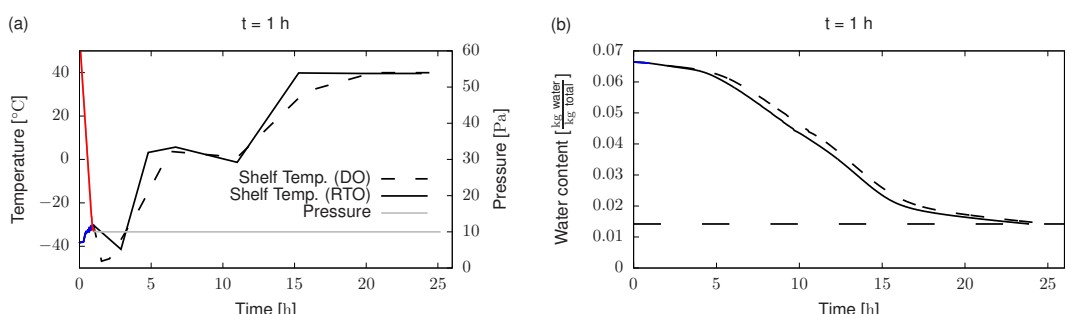

**Figure 7.** Comparison, in terms of process duration, between the RTO approach (continuous black lines) and the off-line control profile (dashed black lines). (**a**) Shelf temperature and chamber pressure. (**b**) Mean water content evolution. Blue and red segments in subplot (**a**) correspond with the real plant measurements for shelf temperature and chamber pressure, respectively.

As shown in the figure, the RTO scheme results into 1.3 h reduction in process duration as compared with the off-line profile. Besides, the RTO scheme would allow to face other two issues [52–55]:

- Unknown disturbances cause lower product temperatures than the predicted ones in the off-line scheme. In this case, final product water content might be larger than the targeted one, i.e., product quality would be lower than required. The RTO scheme can either increase process time or increase shelf temperature to reach the required quality.
- Unknown disturbances cause higher product temperatures than the predicted ones in the off-line scheme. In this case, safety constraints might be not fulfilled which can cause product collapse. The RTO scheme can reduce shelf temperature to avoid product collapse.

## 5. Conclusions and Future Work

In this work, we derived and experimentally validated an operational model describing the behavior of the freeze-drying process. The model focuses on the time scale of the state variables related to product stability and quality, i.e., product temperature and water content. It also describes the evolution of chamber vapor pressure.

Validation results show that the model satisfactorily predicts the evolution of the relevant variables during primary and secondary drying stages. In the transition between these stages minor deviations are found. However, such deviations do not impact the prediction of the product quality. Besides, the model is conservative in the sense that temperature predictions in the transition are larger than the experimental measurements. As a result, control policies derived with the model also satisfy safety constraints in the experiments.

The validated model was subsequently used to derive optimal operation policies aiming at reducing process duration while achieving quality and stability targets. A 30% reduced process duration (13 h) was achieved as compared to standard processes. Finally, we implemented a real-time optimization scheme in which control policies were recomputed on-line using the data measured during the process. This allowed to compensate for unexpected disturbances and model/process mismatch.

The validated operational model and the proposed RTO scheme offer new possibilities to optimize the performance of the freeze-drying process with the only requirement of recalibrating the model for a specific chamber/product.

Future research efforts will be focused on including low level controller (PID) dynamics in the computation of the optimal profiles. In this work, we assume that the low level regulator is able to perfectly follow the optimal profiles so the time required to reach the set point or possible offsets are not taken into account.

**Author Contributions:** Conceptualization, C.V. and A.A.A.; methodology, C.V., A.A.A., E.B.-C. and I.C.T.; software, C.V. and E.L.-Q.; validation, C.V. and I.C.T.; formal analysis, C.V., A.A.A. and I.C.T.; investigation, C.V., A.A.A., I.C.T., E.B.-C. and E.L.-Q.; resources, A.A.A.; writing–original draft preparation, C.V.; writing–review and editing, C.V., A.A.A., E.B.-C., E.L.-Q. and I.C.T.; visualization, C.V.; supervision, C.V. and A.A.A.; project administration, A.A.A.; funding acquisition, A.A.A. All authors have read and agreed to the published version of the manuscript.

**Funding:** This research was funded by E.U. H2020 research and innovation programme (CoPro project, No 723575), E.U. 7th Framework Programme (CAFE Project, KBBE-2007-2-3-01) and Spanish Ministry of Science, Innovation and Universities (InCo4In project, PGC2018-099312-B-C31, MCIU/AEI/FEDER, UE). E. Lopez-Quiroga acknowledges funding received from EPSRC grant no. EP/S023070/1.

**Conflicts of Interest:** The authors declare no conflict of interest.

## Appendix A. The Landau Transform

The well-known Landau transform [34,35] is applied to the Freeze-Drying model presented in Section 2.2 to obtain an equivalent system with a fixed domain. In this section we will present the

model equations after the application of the Landau transform. The following transforms are defined for the dried and frozen regions:

$$Q_{dr} : \xi \to z; \quad \left\{ z \in \mathbb{R} \mid z = \frac{\xi}{x} \right\} \tag{A1}$$

$$Q_{fr} : \xi \to y; \quad \left\{ y \in \mathbb{R} \mid y = \frac{\xi - x}{L - x} \right\} \tag{A2}$$

Note that $z, y \in [0, 1]$. The time coordinate is not transformed although it will be renamed to avoid confusion $\{ \theta \in \mathbb{R} \mid \theta = t \}$ whereas the temperature will be denoted by $\mathcal{T}$, this is: $T_{dr}(t, \xi) \to \mathcal{T}_{dr}(\theta, z)$ and $T_{fr}(t, \xi) \to \mathcal{T}_{fr}(\theta, y)$.

In the new system of coordinates $(z, y)$, Equations (1) and (2) read as:

$$\frac{\partial \mathcal{T}_{dr}}{\partial \theta} = \frac{\alpha_{dr}}{x^2} \frac{\partial^2 \mathcal{T}_{dr}}{\partial z^2} + \frac{zw}{x} \frac{\partial \mathcal{T}_{dr}}{\partial z} \tag{A3}$$

$$\frac{\partial \mathcal{T}_{fr}}{\partial \theta} = \frac{\alpha_{fr}}{(L - x)^2} \frac{\partial^2 \mathcal{T}_{fr}}{\partial y^2} + w \frac{1 - y}{L - x} \frac{\partial \mathcal{T}_{dr}}{\partial y} \tag{A4}$$

where $\alpha_{dr} = \kappa_{dr} / (\rho_{dr} c_{p,dr})$ and $\alpha_{fr} = \kappa_{fr} / (\rho_{fr} c_{p,fr})$ are, respectively, the thermal diffusivity of the dried and frozen regions.

Note that a negative convection term "appears" in the model as a consequence of the application of the transformation. Moreover, two phases are required to apply Stefan condition, see Equation (3), i.e., $0 < x(t) < 1$.

Boundary conditions at the top and the bottom of the sample read as follows in the new coordinate system:

$$\frac{\kappa_{dr}}{x} \frac{\partial \mathcal{T}_{dr}}{\partial z} \bigg|_{z=0} = \sigma e_p f_p (T_{ch}^4 - \mathcal{T}_{dr}|_{z=0}^4) \tag{A5}$$

$$\frac{\kappa_{fr}}{L - x} \frac{\partial \mathcal{T}_{fr}}{\partial y} \bigg|_{y=1} = h_L (T_{sh} - \mathcal{T}_{fr}\big|_{y=1}) \tag{A6}$$

The Stefan Equation (3) reads now:

$$(\rho_{fr} - \rho_{dr}) \Delta H_s w = \left( \frac{\kappa_{fr}}{L - x} \frac{\partial \mathcal{T}_{fr}}{\partial y} \bigg|_{y=0} - \frac{\kappa_{dr}}{x} \frac{\partial \mathcal{T}_{dr}}{\partial z} \bigg|_{z=1} \right) \tag{A7}$$

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
