# Peer review of "Model-Based Real Time Operation of the Freeze-Drying Process"

_processes, doi:10.3390/pr8030325_

Round 1
Reviewer 1 Report
The authors presented the design and validation of model-based real time operation of the freeze-drying process. Based on the theory, a model of freeze-drying process is provided. And the optimizing process is also discussed. However, some concerns are listed as follows:
Since the title mentions about the freeze-drying process, would the authors add the pre-freezing procedure into the model? Table 1 describes most of the parameters utilized in the model. However, there are some parameters in the text are not mentioned. In page 11, the authors mentioned about the mismatch between the predicted and measured temperature. The predicted product temperature is higher than the measured one, meaning that the model will provide less heat than the real need. This may cause the collapse of product. This mismatch may need further discussion. In page 12, the chamber pressure only described about the primary stage. It will be more complete if the transition stage from primary to secondary stage is provided. In introduction, some review about the previous models of freeze-drying process are necessary.Author Response
We thank the reviewers for their effort in revising the manuscript. The thoughtful comments have help to improve the manuscript.
- Since the title mentions about the freeze-drying process, would the authors add the pre-freezing procedure into the model?
The freezing process, which involves crystal formation, is completely different from the drying one. In fact, in some plants, freezing is carried out outside the lyophilization chamber. Modeling this part and including such model in the real time optimization procedure is a subject for a new manuscript and it is out of the scope of this work.
- Table 1 describes most of the parameters utilized in the model. However, there are some parameters in the text are not mentioned.
We have carefully revised the manuscript and included missing parameters in the table. The only parameters not included are those that were grouped in one parameter and therefore, they are not explicitly used in the model. Such parameters are those in Eqs (9) and (13), which are grouped in parameters Kc and beta.
- In page 11, the authors mentioned about the mismatch between the predicted and measured temperature. The predicted product temperature is higher than the measured one, meaning that the model will provide less heat than the real need. This may cause the collapse of product. This mismatch may need further discussion.
There exists some mismatch between measured and predicted product temperature during the transition between primary and secondary drying. However, the model usually predicts a faster increase as compared with the experimental measurements. Product collapse is avoided if product temperature is lower than the glass transition temperature (Eq. 25). Therefore, the model is conservative in the sense that, if the model predicts a product temperature that is lower than the glass transition temperature, then product temperature in the experiments (which is lower than the predicted one) will also be lower than the glass transition temperature.
Paragraph where we discuss this issue has been rephrased to improve clarity (see lines 251-256).
- In page 12, the chamber pressure only described about the primary stage. It will be more complete if the transition stage from primary to secondary stage is provided.
It is true that the sublimation front is not exactly plane (because of heat coming through the edges of the plate) and points closer to the edges would sublimate first. As a consequence, vapor flux and vapor pressure decrease gradually, not instantly. To describe a real physical transition would require a 2D or 3D models, which would require much longer simulation time, and would be unsuitable for real time optimization. Our 1D model cannot describe this transition since there is a single front position so the sublimation flux drops to zero instantly. However, the 1D model represents the worst case scenario, i.e. it would correspond to the slower sublimation point (center of the product) so if safety constraints are fulfilled in this point, they are also fulfilled in the rest of the product.
Besides, during the transition between primary and secondary drying, vapor pressure has little effect on the desorption kinetics so the 1D model can be considered as a good approximation.
We have included a new paragraph (see lines 138-146) to discuss this issue.
- In introduction, some review about the previous models of freeze-drying process are necessary.
Updated references to state of the art models on the freeze-drying of foodstuff have been included in the revised version, third paragraph of the introduction. It must be pointed out that references on modeling of freeze-drying of foodstuff are scarce, most of them focus on the pharmaceutical industry (already included in the previous version).

Reviewer 2 Report
Minor comments:
line 46: optimiZation, not optimisation line 101: "usually" is used two times in the same sentence Please use different layout for plots in Fig.3,5. It is difficult to distinguish between lines even on the screen. In general, it is difficult to work with grey-white-ish lines on the plots. I even did not notice that in Figure 6. Figure 6: sub-figures are not labeled with a,b,c,d,e,f line 336: only one sentence in the paragraph Caption for Figure 3 and lines 229-331 basically give the same information.Major comments:
The authors state that "Previous works on food systems are scarce, and use either lumped inventory models..." (line 41). "... we develop a robust scheme for real time monitoring, control and optimization". However, I found another paper (2004) "Freeze drying process: real time model and optimization" (https://doi.org/10.1016/j.cep.2004.01.005), where authors also use physics based model for real time optimization. It would be good to point out to the differences with the approach suggested, benefits etc. In section 4.2 "Real Time Optimization", the demonstration of the performance of RTO is not convincing. As I understood, black and gray lines in Figure 6a correspond to optimal profile of Temperature and Pressure. Figure 6e shows how these optimal profiles were implemented during the real-time operation. The authors explain the big difference between these profiles by several factors. Particularly, it is stated that (line 322) "Initial chamber pressure was around 60 and it was supposed to be 10..". My question is: should not the initial parameters be controlled by the operator? How can you explain 600% difference in pressure? Authors say that "Despite the issues, the RTO scheme is able to compensate..." (line 326). I expected to see the illustrative demonstration of how the RTO approach outperforms the traditional approach and how it works on practice with real numbers (percent deviation between ideal profile and measured; comparison with traditional approach and improvement in process time in percents). It would be good to show the quantify the differences (average deviation for example) between the experimental data and simulations What are the drawbacks of the suggested model/approach?Author Response
We thank the reviewers for their effort in revising the manuscript. The thoughtful comments have help to improve the manuscript.
Minor comments:
- line 46: optimiZation, not optimisation
We have changed "optimisation" by "optimization" in this line and in others. We have also revised the manuscript to correct similar errors.
- line 101: "usually" is used two times in the same sentence
Second "usually" has been substituted by "often".
- Please use different layout for plots in Fig.3,5. It is difficult to distinguish between lines even on the screen. In general, it is difficult to work with grey-white-ish lines on the plots. I even did not notice that in Figure 6.
The reviewer is right. We have changed the layout of these figures to improve the clarity of the representation.
In Figure 6 we decided not to include the evolution of temperature in the product because it was not necessary for the discussion about the RTO results and would compromise the clarity of the representation. This is why the reviewer could not notice the lines.
- Figure 6: sub-figures are not labeled with a,b,c,d,e,f
In the revised version, labels have been included in this figure.
- line 336: only one sentence in the paragraph
Conclusions (including this paragraph/sentence) have been revised.
- Caption for Figure 3 and lines 229-331 basically give the same information.
We have revised paragraph in lines 229-331 to remove duplicated unnecessary information. We have kept the caption almost unchanged because we think that descriptive captions help the reader to understand the figure. Changes in this caption were motivated by the third comment of this reviewer.
Major comments:
- The authors state that "Previous works on food systems are scarce, and use either lumped inventory models..." (line 41). "... we develop a robust scheme for real time monitoring, control and optimization". However, I found another paper (2004) "Freeze drying process: real time model and optimization" (https://doi.org/10.1016/j.cep.2004.01.005), where authors also use physics based model for real time optimization. It would be good to point out to the differences with the approach suggested, benefits etc.
We were aware of this interesting article, however, honestly, we think the authors do not perform real time optimization. In such paper, the authors developed a 1D model that is able to represent the experimental behavior. Then, they use the model for off-line optimization purposes, i.e. the optimal solution is only computed before the process begins and it is not recomputed as the process evolves using available experimental measurements. In fact, one of the parameters/variables they optimize is the thickness of the product on the tray (L) which does not vary during the process (and cannot be modified). In this regard, and from our point of view, optimization in such work is used for solving a design problem more than a real time optimization problem.
- In section 4.2 "Real Time Optimization", the demonstration of the performance of RTO is not convincing. As I understood, black and gray lines in Figure 6a correspond to optimal profile of Temperature and Pressure. Figure 6e shows how these optimal profiles were implemented during the real-time operation. The authors explain the big difference between these profiles by several factors. Particularly, it is stated that (line 322) "Initial chamber pressure was around 60 and it was supposed to be 10..". My question is: should not the initial parameters be controlled by the operator? How can you explain 600% difference in pressure?
The reviewer is right in that initial parameters should be controlled by the operator. However, sometimes there are issues in the system that do not allow a perfect control of such parameters. This is what happened with the initial product temperature. It was supposed to be -50 Celsius but the refrigerant group was not able to reach such temperature. Regarding initial pressure, the system was able to control it in most of experiments. However, in some of the experiments (in which chamber pressure was fixed to values below 30 Pa) the controller showed difficulties to rapidly keep track of the set points, in particular, at the beginning of the process. In the RTO experiment presented in this work, convergence to 10 Pa was slow. Besides, the controller was not able to maintain such pressure and most of the time chamber pressure was between 5 and 10 Pa. The added value of the RTO scheme is precisely to be able to take into account this kind of unexpected behavior using real plant measurements. We think that this misbehavior in the chamber pressure evolution is good to illustrate the RTO scheme.
- Authors say that "Despite the issues, the RTO scheme is able to compensate..." (line 326). I expected to see the illustrative demonstration of how the RTO approach outperforms the traditional approach and how it works on practice with real numbers (percent deviation between ideal profile and measured; comparison with traditional approach and improvement in process time in percents). It would be good to show the quantify the differences (average deviation for example) between the experimental data and simulations.
Comparison between experimental data and simulations is performed in Section 3, parameter identification. In this section, we use the RMSE value as the objective to minimize. We have also used it in the validation experiments.
Comparison between the traditional approach and the optimal one is performed in section 4.1, off-line dynamic optimization. In this section we showed the advantages of using optimization techniques to obtain the operation conditions. Comparison between the RTO and the traditional approach is difficult to perform in the pilot plant because unexpected disturbances are not reproducible experimentally (they change batch to batch). In any case, a controller that tracks an off-line optimal control can never perform better than an RTO scheme, given the same set of unexpected disturbances. In a plant without disturbances, RTO and off-line optimal control would be the same. As the effect of unexpected disturbances on the process increase, the advantages of RTO become more relevant.
- What are the drawbacks of the suggested model/approach?
The main drawbacks of the approach are: (i) computing optimal policies takes a few minutes, even with the operational model. During this time, although the process continues evolving, plant measurements are not taken into account and the RTO scheme assumes that the plant follows the predicted profiles. Unexpected disturbances during this period can make the profile computed non optimal. However, such disturbances will be taken into account when the next profile is computed. (ii) Low level controller (PID) dynamics are not considered in the computation of the optimal profiles. The approach assumes that the low level regulator is able to perfectly follow the optimal profiles so the time required to reach the set point or possible offsets are not taken into account.
We have discussed these issues in lines 324-328 and the conclusions (lines 372-375).

Round 2
Reviewer 2 Report
In general, I accept the answers to my questions except one.
I would like to get some explanations.
The authors say that:
"Comparison between the traditional approach and the optimal one is performed in section 4.1, off-line dynamic optimization. In this section we showed the advantages of using optimization techniques to obtain the operation conditions. Comparison between the RTO and the traditional approach is difficult to perform in the pilot plant because unexpected disturbances are not reproducible experimentally (they change batch to batch). In any case, a controller that tracks an off-line optimal control can never perform better than an RTO scheme, given the same set of unexpected disturbances. In a plant without disturbances, RTO and off-line optimal control would be the same. As the effect of unexpected disturbances on the process increase, the advantages of RTO become more relevant."
Is there any way to show that RTO scheme outperforms the off-line optimal control? How do we know that RTO does give better results than other approach in the presence of disturbances?
